# Critical role of smoking and household dampness during childhood for adult phlegm and cough: a research example from a prospective cohort study in Great Britain

Noriko Cable,[1] Yvonne Kelly,[1] Mel Bartley,[1] Yuki Sato,[2] Amanda Sacker[1]

[1]International Centre for Life Course Studies in Society and Health (ICLS), Research Department of Epidemiology and Public Health, University College London, London, UK
[2]Centre for Environmental Health Sciences, National Institute for Environmental Studies, Tsukuba-city, Ibaragi, Japan

**Correspondence to**
Dr Noriko Cable;
n.cable@ucl.ac.uk

## ABSTRACT

**Objective:** To examine independent associations between childhood exposures to smoking and household dampness, and phlegm and cough in adulthood.

**Design:** A prospective cohort study.

**Participants:** 7320 of the British cohort who were born during 1 week in 1970 and had complete data for childhood and adult information.

**Main outcome measures:** Experiences of phlegm and coughing over the previous 3 months were assessed using questions from the Medical Research Council (MRC) Questionnaire on respiratory symptoms when the cohort participants were 29 years of age. 4 response patterns (no symptoms, phlegm only, cough only, both symptoms present) were created based on the responses to these questions.

**Results:** Childhood smoking and exposure to marked household dampness at age 10 were associated with phlegm (childhood smoking: relative risk ratio (RRR) =1.45, 95% CI 1.02 to 2.05; dampness: RRR=2.05, 95% CI 1.07 to 3.91) and co-occurring cough and phlegm (childhood smoking: RRR=1.35. 95% CI 1.08 to 1.67; dampness: RRR=2.73, 95% CI 1.88 to 3.99), while exposure to two or more adult smokers in the household was associated with cough-related symptoms (cough only: RRR=1.28, 95% CI 1.04 to 1.58; phlegm and cough: RRR=1.32, 95% CI 1.06 to 1.64). These associations were independent from adult smoking, childhood phlegm and cough, early social background and sex. Current smoking at age 29 contributed to all symptom patterns; however, a substantial association between household dampness and co-occurring phlegm and cough suggest long-term detrimental effects of childhood environmental exposures.

**Conclusions:** Our findings give support to current public health interventions for adult smoking and raise concerns about the long-term effects of a damp home environment on the respiratory health of children.

### Strengths and limitations of this study

- This study uses a large sample with the prospective cohort design, allowing to examine the longitudinal associations between childhood exposures (parental and childhood smoking, household dampness) and adult phlegm and cough without introducing recall bias.
- Confounders (sex, childhood socioeconomic status, respiratory difficulties at birth, childhood phlegm and cough) and adult smoking were accounted to identify significant associations between exposures at age 10 (childhood smoking, parental smoking and household dampness) and adult phlegm and cough at age 29.
- The outcome variable, i.e. patterns of two respiratory symptoms (phlegm and cough), was derived from a well-established questionnaire and not indicative of a particular respiratory disease or lung function.

smoking-related illnesses and mortality.[1] Previous findings have suggested additional risk factors for adult respiratory health. There are longitudinal links between exposure to parental smoking[2–5] or social disadvantage[6–8] during childhood and adult respiratory health, while research evidence to support the association between indoor dampness and respiratory health is mainly cross-sectional.[9–14]

Examining the life course respiratory health of the birth cohort of Britons born in 1946, Mann et al[7] reported that the greatest contribution to adult respiratory health was from proximal risk factors such as adult smoking rather than from distant risk factors such as child respiratory health or social disadvantage. Since adult respiratory health is likely to be determined by child respiratory health,[15] additional longitudinal effects of childhood exposures, such as smoking and

## INTRODUCTION

Smoking cessation has been supported as the main preventive measure to reduce

exposure to household dampness on adult respiratory health may be small.

However, not many studies have used repeated measures of respiratory health in childhood and adulthood. The contribution of childhood exposures to adult respiratory health can be better estimated if a similar respiratory measure is used to assess child and adult respiratory health. We use respiratory symptoms of phlegm and cough occurring in childhood and in adulthood to examine the degree of contribution to these adult respiratory symptoms from exposure to smoking and household dampness during childhood. We used a British cohort who born in 1970 and examined this hypothesis.

Household dampness, exposure to adult cigarette smoke during childhood and childhood smoking will have long-term associations with adult respiratory symptoms over and above childhood respiratory symptoms and smoking in adulthood.

## METHODS
### Study design, setting and participants

We used information obtained from a prospective cohort study, the 1970 British Cohort Study (BCS70). Detailed information about research design and collected data is given by Elliot and Shepherd.[16] BCS70 targets British residents who were born during 1 week in 1970; data have been collected regularly across their life course. We used datasets collected at birth,[17] at age 10[18] and 29[19] to extract study variables. In this study, 7320 available cases which contain all of the study information were used for the analyses.

### Ethical approvals of the study

Data collection at birth and at age 10 was reviewed internally, while ethical approval from the London Multicentre Research Ethics Committee was obtained for the age 29 survey.[20]

### Explanatory factors

We used three childhood exposures (child's own smoking, number of adult smokers in the household, household dampness) as explanatory factors. Information about these factors was collected when the cohort children were 10 years of age. A cohort child's own smoking status was indicated by the response to the question: 'Have you tried a cigarette?' Children who had tried a cigarette were coded as 1 as opposed to those who had not yet experimented with smoking (0). The number of adult smokers in the household was derived through the response about the smoking status of the parents and other adults in the household obtained through a parental interview. This response was tabulated into three categories: 0 for no adult smokers, 1 for one adult and 2 for two or more adults. In our study, a child who smokes could have been exposed to adult smokers at home; however, our interest is to assess the

role of each factor separately, thus we treated these factors independently.

In addition, the degree of household dampness was addressed via this parental interview, asking them to rate the present state of household dampness as: none, slight, moderate or marked. Only a few people responded that their house was 'moderately' damp; therefore, this response was included in the category of 'slight' dampness.

### Adulthood factor

Information about adult smoking was collected when the cohort participants were 29 years of age. Their smoking status was indicated as 0 for current non-smokers and 1 for current smokers.

### Outcome variable

A pattern of co-occurring phlegm and cough (no symptoms, phlegm only, cough only, phlegm and cough) in adulthood was the outcome of this study. Cohort participants were asked to respond whether they usually brought up phlegm or experienced coughing in the morning, during the day or at night in the winter when they were 29 years old. These questions were from the Medical Research Council (MRC) Questionnaire on respiratory symptoms and have been widely used, including in the previous cohort study (ie, National Child Developmental Study).[21] We adapted the method used by Strachan et al[22] and created a variable to indicate presence of each respiratory symptom (phlegm or cough), which cohort participants might have experienced at any time of day. After that we derived a variable with four response patterns of adult respiratory symptoms by combining the response to each variable obtained at age 29: 0 no respiratory symptoms, 1 phlegm only, 2 cough only and 3 phlegm and cough.

### Confounders

We controlled for childhood phlegm and cough, sex and paternal social position. Presence of phlegm (0=no, 1=yes) or cough (0=no, 1=yes) when the cohort children were 10 years old was used. These variables were created through the responses (no, yes but for less than 3 months a year, yes for 3 months or more a year, not known) to the questions: 'Does the study child usually cough/bring up phlegm first thing in the morning/during the day or at night?' These questions are similar to ones from the MRC Questionnaire, with the omission of the phrase 'in the winter'. A positive response to a particular respiratory question was treated as presence of the particular respiratory symptom (phlegm or cough), while the response 'no' was treated as absence of that respiratory symptom. The response 'not known' was regarded as missing and excluded from the analyses. We also included the presence of respiratory difficulties, including respiratory distress syndrome, at the birth of the cohort children (0=absent, 1=present) in the model.

Parental social position was indicated by paternal occupational position at the birth of the cohort child. We adapted the approach by Power et al,[23] classifying the Registrar General's Social Classes of the cohort children's father as: 1—professional or managerial (I and II); 2—skilled non-manual (III non-manual); 3—skilled manual (III manual); 4—partly skilled or unskilled (IV and V) and children who were born in a mother-only household. Midwives provided the information relevant to the birth of the cohort children, while the presence of phlegm or cough was assessed via the parental interview.

## Statistical analysis

Multinomial logistic regression was used to identify associations between explanatory factors and patterns of adult respiratory symptoms (phlegm, cough or both). We took those who did not have any respiratory symptoms as a reference category. First we entered each explanatory variable in separate models, adjusting for sex. In a final model, estimates for all exposures are estimated after adjustment for each other and the confounders. In our study, men and women were analysed together, controlling for the effect of sex. All statistical analyses were performed using Stata SE V.12.[24]

# RESULTS
## Study characteristics

Approximately 2.5% of mothers of the cohort children reported that their children brought up phlegm at age 10, while 11% reported cough (N=7320). Co-occurring symptoms of phlegm and cough was reported for 8.7% of participants at age 29, while 2.9% reported phlegm only and 9.2% cough only. Assessing respiratory symptoms across the life course, only a small proportion of participants continuously experienced respiratory symptoms at both times (table 1).

According to maternal reports, less than half of the cohort children grew up in a smoke-free domestic environment, while household dampness of any degree was observed in around 17% of the homes of the cohort children. Many of the cohort participants did not smoke at 10 years of age, but around half of the cohort participants were current smokers at age 29. Demographically, 48% of the participants were men and 30% of the participants had fathers with non-manual occupations. Most

**Table 1** Description (%) of explanatory and confounding variables by respiratory conditions (no symptoms, phlegm only, cough only, phlegm and cough) at age 29 (N=7320)

| | No symptoms (%) (n=5792) | Phlegm (%) (n=214) | Cough (%) (n=675) | Phlegm+cough (%) (n=639) |
|---|---|---|---|---|
| Explanatory variables | | | | |
| Smoking status at age 10 | | | | |
| Never | 88.09 | 78.97 | 84.74 | 79.50 |
| Smoked | 11.92 | 21.12 | 15.26 | 20.50 |
| Household dampness at age 10 | | | | |
| No dampness | 84.34 | 83.18 | 84.15 | 76.37 |
| Slight to moderate | 13.33 | 11.68 | 12.40 | 16.91 |
| Marked | 2.33 | 5.14 | 3.41 | 6.73 |
| Numbers of adult smokers in the household at age 10 | | | | |
| Zero | 36.08 | 28.50 | 29.93 | 26.60 |
| One | 36.31 | 38.32 | 35.83 | 38.50 |
| Two or more | 27.61 | 33.18 | 34.22 | 34.90 |
| Smoking status at age 29 | | | | |
| Current smoker | 49.64 | 70.56 | 70.81 | 80.44 |
| Confounders | | | | |
| Phlegm at age 10 | | | | |
| Present | 2.05 | 5.61 | 4.00 | 4.38 |
| Cough at age 10 | | | | |
| Present | 9.94 | 14.49 | 14.37 | 16.28 |
| Respiratory difficulties at birth | | | | |
| Present | 2.14 | 2.34 | 2.22 | 2.19 |
| Social position of origin | | | | |
| Professional and managerial | 18.42 | 20.56 | 14.67 | 13.77 |
| Skilled non-manual | 14.50 | 11.21 | 13.04 | 10.33 |
| Skilled manual | 45.51 | 42.99 | 46.96 | 48.36 |
| Non-skilled manual+no male head | 21.56 | 25.23 | 25.33 | 27.54 |
| Sex | | | | |
| Male | 45.98 | 67.76 | 45.63 | 63.85 |
| Female | 54.02 | 32.24 | 54.37 | 36.15 |

of the cohort participants did not experience respiratory difficulties when they were born.

## Long-term associations between childhood factors and adult phlegm or cough

The results of multinomial logistic regression showed childhood factors were independently and specifically associated with patterns of adult respiratory symptoms (phlegm or cough; table 2). The relative risk ratio (RRR) for childhood smoking was 1.45 (95% CI 1.02 to 2.05) for phlegm only and was 1.35 (95% CI 1.08 to 1.67) for co-occurring phlegm and cough. Growing up in a markedly damp house showed a similar, but stronger association with phlegm only (RRR=2.05, 95% CI 1.07 to 3.91) and co-occurring phlegm and cough (RRR=2.73, 95% CI 1.88 to 3.99).

Having adult smokers in the house when the cohort child was 10 years old was positively associated with all patterns of respiratory symptoms; however, having two or more adult smokers were significantly associated with adult cohort participants' respiratory symptoms that were cough only (RRR=1.28, 95% CI 1.04 to 1.58) or co-occurrence of phlegm and cough (RRR=1.32, 95% CI 1.06 to 1.64). These associations were independent of adult smoking and confounders. Adult smoking at age 29 was significantly associated with all of the respiratory response patterns.

## DISCUSSION
### Key results

Using a British cohort born in 1970, we showed significant longitudinal associations between childhood exposures (child's own smoking, two or more adult smokers in the house and marked household dampness) and patterns of adult phlegm or cough. These were independent of presence of phlegm or cough at age 10, adult smoking status, sex and social position of origin. Although the contribution from childhood exposures to adult respiratory symptom patterns (phlegm, cough or both) was relatively small compared to adult smoking, household dampness at age 10 showed a substantial contribution to co-occurring phlegm and cough at age 29.

### Strengths and limitations

This study benefits from a substantially large sample size with a prospective design which limits recall bias. Questions from the MRC Questionnaire were used to identify the patterns of respiratory symptoms (either phlegm or cough, or both symptoms present) at age 29, rather than the presence of respiratory diseases. We consider that identifying respiratory symptoms of phlegm or cough at age 10, using similar questions to the adult questionnaire, enabled us to estimate associations between childhood exposures and adult respiratory symptom patterns (phlegm, cough or both) with considerable accuracy. The omission of the words 'in the winter' in the child version of the respiratory questions,

however, may have overidentified childhood cases of phlegm or cough. In addition, our findings relating to childhood smoking status should be interpreted carefully as they were obtained via self-report. An objective measure of lung function is not available in any sweep of the BCS70 data which is also a limitation to our study.

Information about the duration of exposure to indoor dampness is not available. In their reports published in 2005, the WHO[25] recommends detailed and objective assessment of indoor dampness, such as using a trained assessor or measurements of levels of humidity. In our study, we used parental report on household dampness when the cohort children were 10 years old (ie, 1980). This was the only available measure of household dampness. However, finding a longitudinal association between marked dampness and phlegm related adult respiratory symptoms suggests that exposure to severe indoor dampness may have irritated the respiratory system of the cohort children, even though they were asymptomatic at that time. We are unable to account for air pollution, which could affect indoor air quality via ventilation[25] because the information is not available. Further studies with detailed assessments of household dampness and air quality are needed.

### Possible explanations and comparison with previous studies

We found that experimenting with smoking and exposures to marked household dampness at 10 years of age were longitudinally associated with phlegm-related symptoms at age 29, while having two or more adult smokers in the household showed detrimental associations with cough-related symptoms at the same age. These longitudinal associations are independent of previous phlegm or cough, adult smoking and other confounders. It is worth noting that nearly 80% of the cohort participants (n=11 648) did not have any respiratory symptoms at age 10 to start with, yet 20% of these participants had developed respiratory symptoms, phlegm or cough by age 29. This is somewhat similar to the findings by Guerra et al[3] that lung function became markedly poorer at age 26 only when participants were exposed to both parental smoking at their birth and were smokers themselves. It is possible that development of phlegm or cough may not coincide with earlier exposures to smoking or household dampness. Our findings also add to the existing findings that link parental smoking and respiratory symptoms in adulthood[2 4] and indicate that the mode (active vs second-hand) of childhood exposure to smoking may affect respiratory systems differently.

Our finding linking household dampness with presence of phlegm in adulthood is similar to the findings by Sahlberg et al[26] that adults exposed to household dampness showed increased occurrence of mucosal symptoms 10 years later. Our finding is also similar to the meta-analyses results by Mendell et al[27] showing a substantial association between indoor dampness and various respiratory symptoms, including cough.

**Table 2** Associations between child and adult risk factors and adult respiratory conditions expressed as relative risk ratios (RRR) and 95% CIs in parentheses obtained by multinomial logistic regression (N=7320)

| | No symptoms (n=5792) | Phlegm only (n=214) | | Cough only (n=675) | | Phlegm+cough (n=639) | |
|---|---|---|---|---|---|---|---|
| | | Bivariate* | Fully adjusted† | Bivariate* | Fully adjusted† | Bivariate* | Fully adjusted† |
| Explanatory factors | | | | | | | |
| Smoking status at age 10 (reference: not smoked) | Ref | 1.74 (1.24 to 2.45) | 1.45 (1.02 to 2.05) | 1.34 (1.07 to 1.68) | 1.12 (0.89 to 1.41) | 1.72 (1.40 to 2.13) | 1.35 (1.08 to 1.67) |
| Household dampness at age 10 (reference: no dampness) | | | | | | | |
| Slight to moderate | Ref | 0.89 (0.58 to 1.37) | 0.82 (0.54 to 1.27) | 0.94 (0.74 to 1.19) | 0.85 (0.67 to 1.09) | 1.40 (1.12 to 1.75) | 1.24 (0.99 to 1.56) |
| Marked | Ref | 2.33 (1.23 to 4.41) | 2.05 (1.07 to 3.91) | 1.46 (0.93 to 2.30) | 1.26 (0.80 to 1.99) | 3.30 (2.31 to 4.73) | 2.73 (1.88 to 3.99) |
| Numbers of adult smokers in the household at age 10 (reference: zero) | | | | | | | |
| One | Ref | 1.35 (0.96 to 1.90) | 1.23 (0.88 to 1.74) | 1.19 (0.98 to 1.45) | 1.07 (0.87 to 1.30) | 1.45 (1.18 to 1.78) | 1.20 (0.97 to 1.49) |
| Two or more | Ref | 1.54 (1.09 to 2.19) | 1.36 (0.95 to 1.95) | 1.49 (1.22 to 1.82) | 1.28 (1.04 to 1.58) | 1.74 (1.41 to 2.15) | 1.32 (1.06 to 1.64) |
| Current smoker at age 29 (reference: non-smoking) | Ref | 2.40 (1.78 to 3.24) | 2.29 (1.69 to 3.09) | 2.46 (2.07 to 2.93) | 2.41 (2.02 to 2.88) | 4.13 (3.37 to 5.06) | 3.95 (3.22 to 4.85) |
| Confounders | | | | | | | |
| Phlegm present at age 10 (reference: absent) | Ref | – | 2.62 (1.28 to 5.39) | – | 1.60 (0.99 to 2.58) | – | 1.63 (1.01 to 2.64) |
| Cough present at age 10 (reference: absent) | Ref | – | 1.14 (0.72 to 1.81) | – | 1.34 (1.03 to 1.74) | – | 1.45 (1.12 to 1.89) |
| Sex (reference: men) | Ref | – | 0.42 (0.31 to 0.56) | – | 1.03 (0.87 to 1.21) | – | 0.49 (0.41 to 0.59) |
| Respiratory difficulties at birth (reference: absence) | Ref | – | 1.00 (0.40 to 2.49) | – | 1.06 (0.62 to 1.84) | – | 0.97 (0.54 to 1.72) |
| Social position of origin (reference: professional and managerial) | | | | | | | |
| Skilled non-manual | Ref | – | 0.68 (0.41 to 1.13) | – | 1.14 (0.84 to 1.54) | – | 0.95 (0.68 to 1.33) |
| Skilled manual | Ref | – | 0.81 (0.56 to 1.18) | – | 1.27 (1.00 to 1.62) | – | 1.36 (1.05 to 1.76) |
| Non-skilled manual+no male head | Ref | – | 0.94 (0.61 to 1.42) | – | 1.37 (1.05 to 1.79) | – | 1.46 (1.10 to 1.93) |

*Estimates are adjusted for sex.
†Estimates for explanatory factors are adjusted for sex, confounders and other exposures.

Our finding supports longitudinal associations between childhood exposures and adult respiratory symptoms similar to the study by Mann et al.[7] We are the first to show a substantial contribution of childhood exposure to household dampness to co-occurring phlegm and cough at age 29.

## Implications and conclusions

In summary, we found support for longitudinal associations between childhood exposures to smoking and household dampness, and patterns of adult respiratory symptoms. Although adult smoking contributed to all of adult respiratory symptom patterns, exposure to marked household dampness during childhood showed a substantial contribution to co-occurring phlegm and cough. Our findings support current public health interventions to reduce adult smoking, but also indicate that the management of childhood risk factors such as exposure to smoke (active or second-hand) and household dampness can be a way to prevent adults experiencing poor respiratory health.

**Contributors** NC planned the study, analysed the data and participated in writing and discussion. AS advised statistical analyses and participated in writing and discussion. MB, YK and YS participated in writing and discussion.

**Funding** This work has been funded through the UK Economic and Social Research Council's International Centre for Life Course Studies in Society and Health (ES/J019119/1).

**Competing interests** None.

**Ethics approval** Ethical approval was from the London Multicentre Research Ethics Committee.

**Provenance and peer review** Not commissioned; externally peer reviewed.

**Data sharing statement** BCS70 data used in this study have been managed by the Centre for Longitudinal Study and are accessible via the UK data archive. The statistical code for derived variables is available from the correspondence author.

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
