## [Reviewer comments · BMJ Open]

Some articles will have been accepted based in part or entirely on reviews undertaken for other BMJ Group journals. These will be reproduced where possible.

ARTICLE DETAILS

TITLE (PROVISIONAL)	Critical role of smoking and household dampness during childhood for adult phlegm and cough: A research example from a prospective cohort study in Great Britain
AUTHORS	Cable, Noriko; Bartley, Mel; Kelly, Yvonne; Sato, Yuki; Sacker, Amanda

VERSION 1 - REVIEW

REVIEWER	Dan Norbäck Dept. of Medical Science, Uppsala University, Uppsala, Sweden
REVIEW RETURNED	21-Feb-2014

GENERAL COMMENTS	This is a well-done and important study. I have a few comments and suggestions Title: I think "respiratory health" is too broad, better write "adult phlegm and cough". Same comment for objective better write "adult phlegm and cough". Results; should have some RRR with 95% CI for smoking, Dampness and ETS. I think the Wald Chi-2 analysis can be removed from the abstract, risk estimates is more important. Limitations: You do not measure incidence of cough and phlegm. You could have excluded those with cough and phlegm at 10y and only studied those who had got cough and phlegm during follow-up. Please give a motivation why you have the study design as it is. Introduction OK Material and methods; I understand that active and passive smoking are two separate variables, so a child who is a smoker can still be classified as exposed to passive smoke. Please clarify this. Some other researchers only consider passive smoking as an exposure among non-smokers. Please give more information how the questions on household dampness is asked, and classified. Is it including water leakage, damp spots, window pane condensation, visible mould etc? Please clarify this! What is the time window? Current dampness status or a history of dampness? What is the time window for cough and phlegm at 29, last year, last 2 year or current situation (usually)? Please clarify. Is it the same question for age of 10 and age 29?
---

	Statistical analysis Clarify which factors that were kept in the multinomial regression models. Were all models with mutual adjustment or only one variable with adjustment for sex? Does fully adjusted (in tables) mean mutual adjustment? And what does the Wald Chi-2 analysis add, what is meant here by “overall effect” and is this different from risk estimates from “fully adjusted models”. I think mostly Wald-Chi 2 is used to rank the explanatory value of different variables, but the absolute value of Chi-2 is not so interesting. Results Clarify what the Wald test adds here (see comments above) Discussion Use words cough and phlegm instead of respiratory health (in key results and implications) You should somehow discuss the importance of cough and phlegm within the broader concept of respiratory health, and within this age group (young adults). It would be an indicator of bronchitis, but what has other studies found for the risk factors you have studied in relation to smoking at young age, passive smoking and home dampness? You have one review article on health effects of dampness by Mendell et al, EHP 2011;748-756 that includes cough and phlegm. And you have the WHO dampness and mould document from 2009. Traffic pollution could also give cough and phlegm, could there be a confound by this factor? Do you any information on age of the dwelling at age 10? Old dwellings have more dampness. Clarify what the Wald test adds. It seems dampness is the most important factor, but own smoking gives more phlegm and passive smoking gives more cough, and dampness both Tables Table 2 Clarify what fully adjusted means (give models in a footnote). I guess bivariate does not include adjustment for sex.’ You should clarify social position of origin and what I, II and III means as well as what IV&+ no male head means in a footnote, so the reader do not need to go back to material and methods to find this information. Table 3 clarify what “direct effects” means or use other words.
--	---

REVIEWER	Marzia Simoni Pulmonary Environmental Epidemiology Unit CNR Institute of Clinical Physiology Pisa, Italy
REVIEW RETURNED	24-Feb-2014

GENERAL COMMENTS	-This is an appreciable longitudinal study, that assessed the association between childhood exposure to smoking/household dampness and adult cough/phlegm in a very large sample. The results, besides to confirm the effects of active smoking, confirm the importance of childhood exposure in affecting adulthood respiratory health.
--

	I have very much appreciated this study. It is widely known that early exposure to passive smoking, as well as to mould/dampness, increases the risk of respiratory symptoms in children. However, the author are right when they say most studies are cross-sectional and few studies have used repeated measures of respiratory health. -I have no significant comments to do. Only, I would like to know the question about household dampness exposure. The authors reported the question concerning smoking, but they did not report the question/s concerning home dampness. In general, in studies on indoor pollution, information about presence of mould/damp concern water leakage/water damage indoors in walls/floor/ceiling, bubbles or yellow discoloration on plastic floor covering, black discoloration on parquet floor, visible mould growth on indoors on walls/floor/ceilings, smell of mould in one or more rooms, mould odours in the home, presence of dampness/condensation on the lower part of the windows in winter.... -Page 6, line 30: delete 'g' Table 1. Smoking status: only percent of smoked (current smokers) might be reported; only percent of presence of symptoms might be reported.
--	---

VERSION 1 – AUTHOR RESPONSE

Reviewer(s) Reports:

Reviewer: 1

Reviewer Name Dan Norbäck

Institution and Country Dept. of Medical Science, Uppsala University, Uppsala, Sweden

Please state any competing interests or state 'None declared': None declared

[Comment 1]

This is a well-done and important study. I have a few comments and suggestions

Title: I think "respiratory health" is too broad, better write "adult phlegm and cough".

[Response]

We are delighted by your positive feedback. We took your advice and change the title, abstract, methods and findings accordingly.

[Comment 2]

Same comment for objective better write "adult phlegm and cough".

[Response] Please refer to the response to the previous comment.

[Comment 3]

Results; should have some RRR with 95% CI for smoking, Dampness and ETS. I think the Wald Chi-2 analysis can be removed from the abstract, risk estimates is more important.

[Response] We included RRR results and removed results from the Wald test.

[Comment 4]

Limitations: You do not measure incidence of cough and phlegm. You could have excluded those with cough and phlegm at 10y and only studied those who had got cough and phlegm during follow-up.

Please give a motivation why you have the study design as it is.

[Response] We added a sentence, 'Current smoking at age 29 contributed to all symptom patterns; however, a substantial association between household dampness and co-occurring phlegm and cough suggest long-term detrimental effects of childhood environmental exposures'. (Results under Abstract) It ties up into our objective to examine independent associations between childhood exposures to smoking and housing dampness and phlegm and cough in adulthood.

Introduction OK

[Comment 5]

Material and methods; I understand that active and passive smoking are two separate variables, so a child who is a smoker can still be classified as exposed to passive smoke. Please clarify this. Some other researchers only consider passive smoking as an exposure among non-smokers.

[Response]

The referee is right that we independently estimated the effects from active and passive smoking during childhood on adult respiratory symptom patterns. We added a sentence in the method section to state our motivation for this: 'In our study a smoking child could have been exposed to adult smokers at home; however our interest is to assess the role of each factor separately, thus we treated these factors independently'. (Explanatory factors, under Method).

[Comment 6]

Please give more information how the questions on household dampness is asked, and classified. Is it including water leakage, damp spots, window pane condensation, visible mould etc? Please clarify this! What is the time window? Current dampness status or a history of dampness?

[Response]

We added 'present state of' to household dampness. (Explanatory factors under Method) This was assessed by a parental response to the question asking about a degree of household dampness. It was not detailed as this referee asked. This was the only information collected when the survey was taken place (cohort children at age 10, 1980). However we acknowledge this as limitation and addressed in the limitation section.

[Comment 7]

What is the time window for cough and phlegm at 29, last year, last 2 year or current situation (usually)? Please clarify. Is it the same question for age of 10 and age 29?

[Response]

In original manuscript cohort participants were asked to report whether they experienced bringing up phlegm or experienced coughing in the morning, during the day or at night in the winter when they were 29 years old. The actual question included 'usually'; we added it to be more accurate. Thank you very much for this feedback.

In relation to age 10 respiratory symptom question, the wording was similar. The word 'in the winter' was excluded for the childhood respiratory symptom question. It is described in the original manuscript. (Confounders, under Method)

[Comment 8]

Statistical analysis

Clarify which factors that were kept in the multinomial regression models. Were all models with mutual adjustment or only one variable with adjustment for sex? Does fully adjusted (in tables) mean mutual adjustment? And what does the Wald Chi-2 analysis add, what is meant here by "overall effect" and is this different from risk estimates from "fully adjusted models". I think mostly Wald-Chi 2 is used to rank the explanatory value of different variables, but the absolute value of Chi-2 is not so

interesting.

[Response]

We appreciate the referee's valuable comment. We revised the statistical analyses section to reflect this feedback. It now reads: 'First we entered each explanatory variable in separate models, adjusting for sex. In a final model estimates for all exposures are estimated after adjustment for each other and the confounders'. (Statistical analysis)

We reflected on the comments from the referees 1 and 2 regarding the results of the Wald test. We concluded that the results did not add to the findings. Point estimates are more important in this study. We deleted all results from Wald test from the manuscript not to confuse readers.

[Comment9]

Results

Clarify what the Wald test adds here (see comments above)

[Response]

Please refer to the response above.

[Comment10]

Discussion

Use words cough and phlegm instead of respiratory health (in key results and implications)
You should somehow discuss the importance of cough and phlegm within the broader concept of respiratory health, and within this age group (young adults). It would be an indicator of bronchitis, but what has other studies found for the risk factors you have studied in relation to smoking at young age, passive smoking and home dampness? You have one review article on health effects of dampness by Mendell et al, EHP 2011;748-756 that includes cough and phlegm. And you have the WHO dampness and mould document from 2009. Traffic pollution could also give cough and phlegm, could there be a confound by this factor? Do you any information on age of the dwelling at age 10? Old dwellings have more dampness.

[Response]

Thank you again for pointing us about excellent references.

First of all we changed from 'respiratory health' to either 'phlegm or cough' or 'respiratory symptoms' where appropriate. (Discussion)

We included two references to elaborate our discussion. The information about household dampness is based on one question about dampness of the present household. There were no questions about traffic pollution, either. As we described in the response to the comment 7, we discussed this methodological limitation referring to this WHO reference. Please note that child respiratory questions were asked in 1980 when the cohort children were 10 year old. We suggested a need for further investigation with detailed assessment tools. (Discussion)

[Comment10]

Clarify what the Wald test adds. It seems dampness is the most important factor, but own smoking gives more phlegm and passive smoking gives more cough, and dampness both

[Response]

We took your valuable feedback on board. We discussed and concluded that the results from the Wald test do not add anything to the estimates from multinomial logistic regression. We therefore removed texts mentioning about this from the manuscript.

[Comment11]

Tables

Table 2 Clarify what fully adjusted means (give models in a footnote). I guess bivariate does not include adjustment for sex.'

[Response]

We apologise for not being explicit about our modelling strategies when we originally presented the table 2. The referee was right about bivariate results should be adjusted for sex. We now present bivariate estimates which were adjusted for sex. Fully adjusted means estimates were adjusting for other explanatory factors and confounders. We added a footnote to describe each model (table 2).

[Comment12]

You should clarify social position of origin and what I, II and III means as well as what IV&+ no male head means in a footnote, so the reader do not need to go back to material and methods to find this information.

[Response]

We sincerely apologise for not being explicit about the classification of occupational position in the UK. We added the labels in the table accordingly. It now reads: Professional and managerial, Skilled non-manual, Skilled manual, Non-skilled manual + no male head (Tables 1 and 2).

[Comment13]

Table 3 clarify what “direct effects” means or use other words.

[Response]

It should be read ‘total’ effects; however, we agreed that the results did not add to our findings. In the light of your comment we concluded having the results from the Wald test is confusing. Therefore we decided to remove the table 3. We however can offer the table upon request.

Reviewer: 2

Reviewer Name Marzia Simoni

Institution and Country Pulmonary Environmental Epidemiology Unit

CNR Institute of Clinical Physiology

Pisa, Italy

Please state any competing interests or state ‘None declared’: None declared

[Comment 1]

This is an appreciable longitudinal study, that assessed the association between childhood exposure to smoking/household dampness and adult cough/phlegm in a very large sample. The results, besides to confirm the effects of active smoking, confirm the importance of childhood exposure in affecting adulthood respiratory health.

I have very much appreciated this study. It is widely known that early exposure to passive smoking, as well as to mould/dampness, increases the risk of respiratory symptoms in children. However, the author are right when they say most studies are cross-sectional and few studies have used repeated measures of respiratory health.

I have no significant comments to do.

Only, I would like to know the question about household dampness exposure.

The authors reported the question concerning smoking, but they did not report the question/s concerning home dampness. In general, in studies on indoor pollution, information about presence of mould/damp concern water leakage/water damage indoors in walls/floor/ceiling, bubbles or yellow discoloration on plastic floor covering, black discoloration on parquet floor, visible mould growth on indoors on walls/floor/ceilings, smell of mould in one or more rooms, mould odours in the home, presence of dampness/condensation on the lower part of the windows in winter....

[Response]

Thank you very much for your feedback. As we responded to the referee 1’s comment (see response to the comment 5), we described how household dampness was assessed in the original manuscript. We made it more explicitly. We added the time framework of household dampness by adding the phrase of ‘the present state of’. It now reads ‘In addition the degree of household dampness was

addressed via this parental interview, asking them to rate the present state of household dampness as: none, slight moderate or marked'.

We acknowledged the limitation to our method for dampness measure and it was addressed in the discussion section. ('Strengths and limitations' under discussion)

[Comment2]

Page 6, line 30: delete 'g'

[Response] Thank you for spotting this typo. We deleted the letter from the manuscript.

[Comment3]

Table 1. Smoking status: only percent of smoked (current smokers) might be reported; only percent of presence of symptoms might be reported.

[Response] We amended our reports on Table 1 accordingly.